# A New Era in Endothelial Injury Syndromes: Toxicity of CAR-T Cells and the Role of Immunity

**DOI:** 10.3390/ijms21113886

**Published:** 2020-05-29

**Authors:** Eleni Gavriilaki, Ioanna Sakellari, Maria Gavriilaki, Achilles Anagnostopoulos

**Affiliations:** 1Hematology Department—BMT Unit, G Papanicolaou Hospital, 57010 Thessaloniki, Greece; ioannamarilena@gmail.com (I.S.); achanag@gmail.com (A.A.); 2Laboratory of Clinical Neurophysiology, AHEPA Hospital, Aristotle University of Thessaloniki, 54636 Thessaloniki, Greece; mariagavri6@yahoo.gr

**Keywords:** CAR-T, endothelial injury syndrome, toxicity, complement, immunity

## Abstract

Immunotherapy with chimeric antigen receptor T (CAR-T cells) has been recently approved for patients with relapsed/refractory B-lymphoproliferative neoplasms. Along with great efficacy in patients with poor prognosis, CAR-T cells have been also linked with novel toxicities in a significant portion of patients. Cytokine release syndrome (CRS) and neurotoxicity present with unique clinical phenotypes that have not been previously observed. Nevertheless, they share similar characteristics with endothelial injury syndromes developing post hematopoietic cell transplantation (HCT). Evolution in complement therapeutics has attracted renewed interest in these life-threatening syndromes, primarily concerning transplant-associated thrombotic microangiopathy (TA-TMA). The immune system emerges as a key player not only mediating cytokine responses but potentially contributing to endothelial injury in CAR-T cell toxicity. The interplay between complement, endothelial dysfunction, hypercoagulability, and inflammation seems to be a common denominator in these syndromes. As the indications for CAR-T cells and patient populations expand, there in an unmet clinical need of better understanding of the pathophysiology of CAR-T cell toxicity. Therefore, this review aims to provide state-of-the-art knowledge on cellular therapies in clinical practice (indications and toxicities), endothelial injury syndromes and immunity, as well as potential therapeutic targets.

## 1. Introduction

Both autologous and allogeneic hematopoietic cell transplantation (HCT) have been widely applied for the treatment of hematologic and autoimmune diseases [1,2]. Autologous HCT provides the opportunity of intensive chemotherapy and immunosuppression, whereas allogeneic HCT provides additional benefits of anti-tumor effects through immune mechanisms [3]. A novel cellular therapy in the field of autologous HCT has been recently approved in patients with hematologic malignancies: immunotherapy with chimeric antigen receptor T (CAR-T cells). Along with great efficacy in patients with poor prognosis, CAR-T cells have been also linked with novel toxicities in a significant portion of patients [4,5,6].

Despite efforts to characterize this new toxicity profile, our understanding of the pathophysiology and potential therapeutic targets remains poor. These syndromes resemble the endothelial injury syndromes observed post allogeneic HCT [7], but present with a different phenotype. Therefore, this review aims to provide state-of-the art knowledge on cellular therapies in clinical practice (indications and toxicities), endothelial injury syndromes and immunity, as well as potential therapeutic targets.

## 2. Cellular Therapies in Clinical Practice

### 2.1. Indications

Recently, two CAR-T cell products have been approved for use in clinical practice:(1)Tisagenlecleucel (KYMRIAH, Novartis, Basel, Switzerland) is a biosynthetic CD19 CAR-T cell product, approved for treatment of children and young adults (up to 25 years of age) suffering from relapsed/refractory B acute lymphoblastic leukemia (ALL) [4], as well as certain types of relapsed/refractory aggressive B non-Hodgkin lymphoma (NHL) [5].(2)Axicabtagene ciloleucel (YESCARTA, Kite Pharma, a Gilead Company, Los Angeles, CA, USA) is also a biosynthetic CD19 CAR-T cell product, approved for treatment of certain types of relapsed/refractory aggressive B non-Hodgkin lymphoma (NHL) [6].

Other CAR-T cell products have also shown promising results [8]. In particular, Lisocabtagene maraleucel (Liso-cel, Bristol Myers Squibb) is undergoing a priority review for relapsed/refractory large B-cell lymphoma.

Manufacturing these CAR-T cell products has been the result of continuous research in the field since 1993. This research has moved the field from T-cell receptor mimetics to fourth generation CARs [9]. Briefly, first generation CARs include an scFv antigen-binding epitope with one signaling domain. The CD3ζ chain provides signals required for T cell activation. In second generation, a costimulatory molecule, mainly CD28 or 4-1BB receptor (CD137), is added. The approved products that have been mentioned above are of second generation. Third generation CARs improve effector functions and persistence compared to second generation. Finally, fourth generation CARs are also called TRUCKs (CAR redirected T cells that deliver a transgenic product to the targeted tumor tissue) or armored CARs. These present enhanced antitumor potency, cytokine activity, and costimulatory ligands [10].

Except for the construct, the success of CAR-T cell lies in the selection of an optimal cell surface antigen as a target. CD19 has been selected as an optimal target for several reasons. It is expressed in the cell surface primarily of the B-cell lineage, with highly restricted expression in normal tissues [11]. It is also involved in B-cell development and function, and possibly in tumor biology [9].

The process of CAR-T administration to patients resembles that of autologous HCT [2]. This autologous process requires leukapheresis of selected patients. T cells are then isolated and genetically engineered to express a modified T cell receptor. CAR-T cells are subsequently infused to the patient after a lymphodepleting regimen [12]. Patients need to be carefully monitored for toxicities and therefore, CAR-T cell therapy is currently performed in accredited transplant units. Interestingly, successful outpatient treatment is currently performed with new products, such as liso-cel [13].

The above-mentioned indications of CAR-T cell products were based on their efficacy in patients with otherwise poor outcomes. Indeed, relapse is observed in up to 15–20% of children and young adults with ALL, with an overall survival of 22% at 1 year and 7% in 5 years. In relevant clinical trials of CAR-T cells, response reached 81% [4]. Similarly, patients with relapsed/refractory aggressive B-non-Hodgkin Lymphoma (B-NHL) do not benefit from autologous HCT, since relapse is observed in 60% of those that undergo autologous HCT. CAR-T cells showed a complete remission of 38% in clinical trials [14], with similar efficacy in real-world data [15].

### 2.2. Toxicity

CAR-T cells have introduced a novel toxicity paradigm. Clinical manifestations vary and affect multiple systems, as summarized in Table 1. Cardiac, respiratory, hepatic, or gastrointestinal, hematologic and renal toxicity are usually reversible or transient. Mild symptoms from these systems, such as cough or nausea, are very common (≥1/10); while severe events, such as infarcts or B-cell aplasia, uncommon (≥1/1000 to <1/100). Among them, two syndromes require intensive management: cytokine release syndrome (CRS) and neurotoxicity, recently re-named to Immune effector cell-associated neurotoxicity syndrome (ICANS). In the phase 3 trials of approved products (including liso-cel), CRS was observed in 37–93% of patients, and neurotoxicity in 23–65% [4,6,16]. CRS manifests with fever, hypotension, hypoxia, manifestations from multiple systems: arrhythmia, cardiomyopathy, prolonged QTc, heart block, renal failure, pleural effusions, transaminitis, and coagulopathy. Neurotoxicity may present with delirium, encephalopathy, somnolence, obtundation, cognitive disturbance, dysphasia, tremor, ataxia, myoclonus, focal motor and sensory defect, seizures, cerebral edema. Given the diverse clinical manifestations of these syndromes, increased awareness is needed to early diagnose them post CAR-T cell therapy.

Data from different clinical studies and research groups have been recently harmonized based on a consensus grading suggested by the ASCT (American Society of Transplantation and Cellular Therapy, formerly American Society for Blood and Marrow Transplantation, ASBMT) [17]. This consensus document provides additional useful recommendations on tools to diagnose and monitor patients. Although these syndromes have been reported in up to 93% of patients early after CAR-T cell therapies, they are potentially life-threatening [18].

Management is also based on a multidisciplinary approach, with a significant portion of patients in need of intensive care and neurology consultation. The use of steroids and tocilizumab, an IL-6 agent, seem to mainly abrogate CRS and subsequently, neurotoxicity, since these toxicities commonly co-exist. Although neurotoxicity is reversible in most cases, 3–10% of neurologic events remain unresolved [19]. Importantly, deaths due to toxicity syndromes have been reported, despite optimal management [18].

Despite increased interest in these toxicities, their pathophysiology has not been clarified yet [20]. Hunter and Jacobson have recently reviewed the pathophysiology focusing on neurotoxicity [19]. As explicitly shown in their review, no experimental model or in vitro study has so far replicated the profile of neurotoxicity post CAR-T cells. It is widely accepted that these syndromes are characterized by endothelial injury and hypercoagulability [18]. Although the latter are common denominators in well described endothelial injury syndromes post allogeneic HCT [7], no direct link has yet been established.

## 3. Endothelial Injury Syndromes and Immunity

### 3.1. Endothelial Injury Syndromes

Various endothelial injury syndromes result post allogeneic HCT, including transplant- associated thrombotic microangiopathy (TA-TMA), graft-versus-host disease (GVHD) and veno-occlusive disease/sinusoidal obstruction syndrome (SOS/VOD) [21].

TA-TMA is a life-threatening complication of HCT that manifests with microangiopathic hemolytic anemia, thrombocytopenia and often renal or neurologic dysfunction [22,23,24,25,26,27]. It is more common post allogeneic HCT, but has also been described post autologous HCT, especially in pediatric recipients [28]. Its diagnosis is largely hindered by the high incidence of cytopenias and organ dysfunction in HCT recipients. Indeed, renal and neurologic dysfunction are attributed to several causes post HCT, that are potentially life-threatening [29,30,31]. Endothelial injury has been long recognized as a contributor to the pathogenesis of TA-TMA. Various underlying processes (conditioning regimen toxicity, calcineurin inhibitors/CNIs, alloreactivity, bacterial products, and GVHD) contribute to a prothrombotic state, which may eventually lead to microvasculature thrombosis [32].

GVHD is the major cause of morbidity and mortality among allogeneic HCT survivors without relapse or secondary malignancy [33,34]. GVHD treatment consists mainly of immunosuppressive agents [35]. Prolonged immunosuppression is a risk factor of severe infections, leading to a vicious cycle of morbidity in GVHD patients [30,36,37]. Markers of endothelial dysfunction, such as endothelial microvesicles [38], are significantly increased 2–3 weeks post allogeneic HCT [39], as well as in patients with acute GVHD [40]. Endothelial activation has also been implicated in the pathophysiology of acute GVHD by a recent experimental study [41].

SOS/VOD disease of the liver has been traditionally considered a severe complication of allogeneic HCT, particularly in patients with known risk factors [42]. Although it manifests as a rare HCT complication thanks to advances in transplant modalities [43,44], calicheamicin-conjugated antibodies, gemtuzumab and inotuzumab ozogamicin, have led to renewed interest in this syndrome [45,46]. Our group along with others has shown that changes in coagulation and fibrinolysis are predictive of SOS/VOD [47]. However, further studies have failed to identify useful biomarkers for routine clinical practice [42]. Its pathophysiology is strongly associated with damage observed in sinusoidal endothelial cells and in hepatocytes that continues with progressive venular occlusion [42].

Recent progress, mainly in the field of thrombotic microangiopathies (TMAs), has highlighted the role of complement as a common denominator in endothelial injury syndromes [48].

### 3.2. Immunity

The complement system is part of the immune system, comprising of more than 50 soluble and membrane-bound proteins [49]. It provides innate defense against microbes and mediating inflammatory responses. Except for inflammation, a link also exists between the complement system and platelet activation, leukocyte recruitment, endothelial cell activation and coagulation. Several reviews have tried to delineate the complex link between complement and thrombosis [50,51]. This link is basically established through interactions between C3, C5, and thrombin. Figure 1 summarizes complement activation and its interaction of complement with other pathways, that may implicate it in CAR-T cell toxicity.

The proximal complement cascade is activated by the classical, alternative, and lectin pathways. The classical pathway is mainly activated by antibody-antigen complexes recognized by complement component C1q [52]. This leads to the formation of classical pathway C3 convertase that cleaves C3, generating the anaphylatoxin C5a and C5 convertase. The latter cleaves C5 into C5a and C5b, initiating the terminal pathway of complement. In the terminal pathway, C5b binds to C6 and C7 generating C5b-7, that is able to insert into lipid layers of the membrane [53]. C5b-7 binds C8 and C9, forming a complex that unfolds in the membrane and binds several C9 molecules, thereby forming the membrane attack complex (MAC).

Interestingly, the alternative pathway of complement serves as an amplification loop for the lectin and classical pathways, accounting for roughly 80% of complement activation products [54]. The alternative pathway is continuously activated through slow spontaneous hydrolysis of C3, which forms C3(H_2_O) [55]. The activated C3(H_2_O) binds factor B, generating C3(H_2_O)B. Factor B is subsequently cleaved by factor D, generating the fluid phase APC C3 convertase, or C3(H2O)Bb. C3 convertase then catalyzes the cleavage of additional C3 molecules to generate C3a and C3b, which attach to cell surfaces [56]. This initiates the amplification loop, where C3b pairs with factor B on cell surfaces, bound factor B is cleaved by factor D to generate a second APC C3 convertase (C3bBb). Membrane-bound C3 convertase then cleaves additional C3 to generate more C3b deposits, closing the amplification loop. The binding and cleavage of an additional C3 molecule to C3 convertase forms the C5 convertase, initiating terminal pathway activation. Both C3 and C5 APC convertases are stabilized by properdin [57], which also serves as a selective pattern recognition molecule for de novo C3 APC convertase assembly [55]. Properdin is the only known positive regulator of complement. It increases the activity of C3 and C5 convertases, which amplify C3b deposition on cell surfaces [58].

Lectin pathway activation is initiated by mannose-binding lectins (MBLs) [59,60] and other pattern recognition molecules including ficolins and collectin 11 [61]. These molecules act through MBL-associated serine proteases (MASPs), which generate the C3 convertase in a process similar to that of the classical pathway.

### 3.3. Complement Activation in Endothelial Injury Syndromes

In TA-TMA, Jodele et al. first suggested that TA-TMA results from endothelial dysfunction after multiple triggers in genetically predisposed pediatric patients [27,62]. Initial data have shown excessive activation of terminal complement pathway through a rough marker of terminal complement activation, soluble C5b-9 levels [27]. Further studies have also confirmed complement activation on cell surface through functional assays [63]. Additionally, genomic data have suggested genetic susceptibility through rare mutations in complement-mediated genes [62]. Our group confirmed these data in adult patients [64], providing additional evidence of a vicious cycle of endothelial dysfunction, hypercoagulability, neutrophil and complement activation in TA-TMA [7]. A more recent study of transcriptome analysis in pediatric TA-TMA has shown activation of multiple complement pathways and an interplay between complement and interferon that perpetuates endothelial injury [65]. These data are in line with a previous clinical observation documenting complement-mediated TMA in patients with hemophagocytic lymphohistiocytosis (HLH), a rare clinical syndrome of excessive immune activation, characterized by signs and symptoms of extreme inflammation, largely driven by interferon γ and other pro-inflammatory cytokines [66].

Our understanding of the pathophysiology of TA-TMA has led to a revolution in therapeutics. Based on their success in patients with TMA and excessive complement activation [67,68], complement inhibitors have also shown success in TA-TMA. The first-in-class terminal complement inhibitor, eculizumab, has long been used in TA-TMA [69,70,71,72]. Real-world data suggest early initiation of treatment in patients with complement activation measured by soluble C5b-9 levels, as well as monitoring of treatment and dose adjustments yield better results [73]. Recently, narsoplimab (OMS721), a novel lectin pathway inhibitor targeting MASP-2 (mannan-binding lectin-associated serine protease-2), received breakthrough FDA designation, based on positive data in TA-TMA [74].

Clinical features of SOS/VOD share common characteristics with a syndrome observed during pregnancy, the HELLP (hemolysis, elevated liver enzymes, and low platelet number) syndrome. We and other groups have provided functional and genetic evidence pointing towards increased complement activation associated with complement-related germline mutations in patients with HELLP syndrome [75,76,77,78,79]. In this context, these syndromes resemble the disease model of complement-mediated hemolytic uremic syndrome (HUS) [48]. Different mutations in complement- related factors may lead to distinct phenotypes with similar characteristics as shown in other complement-related diseases, such as C3G-glomerupathy and age-related macular degeneration [80,81].

Earlier studies have suggested preliminary evidence of complement activation in patients with SOS/VOD. A subset of transplanted patients with SOS/VOD has shown increased complement activation markers at levels similar to those of patients with transplant-associated TMA. In addition, ADAMTS13 (A Disintegrin and Metalloproteinase with Thrombospondin motifs), a known regulator of TMAs, was reported lower in patients with SOS/VOD [82]. In line with these data, a previous case report documented increased complement activation in a SOS/VOD patient that was efficiently treated with the complement inhibitor C1 esterase inhibitor (C1-INH-C) [83]. Regarding genetic studies, Bucalossi et al. detected two complement factor H (CFH) variants in 3 SOS/VOD patients. Except for complement factor I (CFI), no other complement-related genes were studied [84].

## 4. CAR-T Cell Toxicity and Endothelial Injury Syndromes

Endothelial dysfunction and hypercoagulability are being currently investigated in CAR-T cell toxicity. Indeed, angiopoietin 2, the angiopoietin-2 to angiopetin-1 ratio, and von Willebrand Factor (VWF) were increased in patients with severe neurotoxicity (grade ≥ 4) [85]. These patients also had a lower fraction of high molecular weight VWF multimers and a higher fraction of low molecular weight VWF multimers. In addition, ADAMTS13 that cleaves VWF [86], has been measured in patients with severe neurotoxicity. A lower ADAMTS13:VWF ratio was found when compared to patients with lower grade neurotoxicity [85]. To further confirm evidence of endothelial activation, sera from patients with severe neurotoxicity induced the formation of VWF-platelet strings on endothelial cells in vitro [85]. Similarly, high angiopoietin-2 to angiopoietin-1 ratios were found in patients with severe neurotoxicity, grade 3–4 compared to grade 0–2 [87]. Angiopoietin-2 and VWF were also increased during severe CRS and before lymphodepletion in patients who developed CRS [88].

Additional findings of hypercoagulability have been suggested by laboratory markers of disseminated intravascular coagulation (DIC) in these patients [87]. This evidence is tightly linked to blood-cerebrospinal fluid barrier disruption and proinflammatory cytokines that have been also observed in severe neurotoxicity [85,87]. Endothelial expression of adhesion molecules (ICAM-1 and VCAM-1) has also been found impaired in experimental studies [89].

An alternative link between endothelial injury and CAR-T cell toxicity stems from cardiovascular events. Although cardiovascular events are largely under-reported in patients with hematologic malignancies and cellular therapies [32], a recent study in patients post CAR-T cell therapy has highlighted their role. In particular, cardiac injury and cardiovascular events were common, showing a graded relationship among CRS, elevated troponin, and cardiovascular events. A lower rate of cardiovascular events was found in CRS patients with early onset of tocilizumab [90]. It should be noted that endothelial dysfunction is considered an early event in the pathophysiology of cardiovascular disease [91]. In this context, CAR-T cell toxicity resembles endothelial injury syndromes.

Finally, a unique characteristic of CAR-T cell toxicity is the central role of the immune system orchestrating the cytokine storm. Monocytes and macrophages are key cells in this process. Giavridis et al. documented that CRS severity depends on IL-6, IL-1, and nitric oxide produced by macrophages in mice models [92]. Even in neurotoxicity, macrophage infiltration in the subarachnoid space has been shown in animal studies [93] and in a patient with a fatal outcome [94].

## 5. Potential Therapeutic Targets in CAR-T Cell Toxicity

Supportive care, corticosteroids, and tocilizumab are the mainstay of treatment for the potentially life-threatening CAR-T cell toxicity [95]. Innovative approaches have focused on developing alternative products linked with lower toxicity rates. A really interesting study utilized low molecular weight adapters to regulate toxicity post CAR-T cell therapy [96]. Other approaches targeting IL-1 through competitive binding to IL-1 receptor (anakinra) or anti- inflammatory cytokines (IL-37) have also been suggested [97]. Interestingly, anakinra decreased CRS and neurotoxicity-mediated mortality in mice studies [92,94].

Nevertheless, anti-inflammatory approaches address rather the pathophysiology of cytokine storm and not that of endothelial injury and neurotoxicity. Therefore, strategies addressing the endothelial activation and related pathways may be adapted from successful treatment of endothelial dysfunction syndromes. Defibrotide is the only efficient treatment of SOS/VOD, with encouraging results as a prophylactic treatment in high-risk patients [98,99]. Defibrotide which dramatically improves survival in patients with SOS/VOD through endothelial stabilization, may work in a similar manner in prevention or treatment of neurotoxicity following CAR T-cell therapy [42].

### 5.1. Complement Inhibition

Complement inhibition is the treatment of choice for several complement-mediated diseases [49]. The disease model for complement inhibition is paroxysmal nocturnal hemoglobinuria (PNH), patients manifesting with severe hemolytic anemia and/or thrombosis. Two complement inhibitors have been approved by the FDA for the treatment of PNHs: eculizumab in 2007 and ravulizumab in 2019. Both monoclonal antibodies are administered intravenously. They block terminal complement activation by binding to C5 and sterically hindering cleavage of C5 by the C5 convertase. As a result, the generation of the proinflammatory C5a molecule and MAC formation are blocked [100,101]. Ravulizumab has the advantage of 4-fold longer half-life, showing a non-inferior efficacy compared to eculizumab in large PNH trials [102,103]. Recently, ravulizumab has shown sustained one-year safety and efficacy [104], as well as decreased breakthrough hemolysis [105]. Ravulizumab is currently considered the drug of choice based on its long half-life that allows for more convenient dosing. The only major adverse effect from terminal complement inhibition has been an expected increase risk of neisserial meningitis (0.5% risk annually), since infection for other encapsulated bacteria is of lower risk [106]. The predictable toxicity from C5 inhibition and lack of other major end- organ toxicity along with a life-changing efficacy, have rendered complement inhibition a precision medicine paradigm.

Novel complement inhibitors are in the advanced phase of clinical development [107,108,109]. Providing further details does not fall within the scope of this review. Therefore, we will briefly describe the advantages and disadvantages of agents already administered in the disease model of complement activation, PNH. The terminal complement inhibitor crovalimab also targets C5, but at a different epitope from eculizumab and ravulizumab. It has the advantage of subcutaneous administration every four weeks [110]. Proximal complement inhibition targets complement proteins upstream of CD59 and CD55, such as C3, factor D, and factor B. These inhibitors are expected to be more precise, especially in PNH. The factor D inhibitor (danicopan) has the advantage of oral administration [111,112]. Regarding infection concerns, experimental studies have shown that danicopan preserves the activity of classical and lectin pathways against invasive pathogens [113]. Furthermore, increased meningococcal killing in vaccinated volunteers has been shown in the presence of danicopan compared to anti-C5 inhibitors [114]. Pegcetacoplan is administered subcutaneously. This 15-amino acid cyclic peptide is conjugated to polyethylene glycol. It binds to C3 and prevents C3 and C5 cleavage by their respective convertases [115]. In summary, the therapeutic armamentarium of complement-mediated diseases is constantly expanding. The choice of the ideal complement inhibitor is soon expected to be personalized.

### 5.2. Complement Inhibition in Neurotoxicity

There is no clear evidence of complement inhibitor actions against neurotoxicity. Interestingly, complement inhibition has been recently approved for certain neurological disorders. In addition, similar to the mechanisms of actions in endothelial injury syndromes, complement inhibitors are expected to provide benefits against endothelial dysfunction. Importantly, complement inhibitors are also expected to induce an anti-inflammatory effect. The continuous progress made during the recent coronavirus (COVID-19) pandemic has provided important insight into the anti-inflammatory effects of complement inhibitors.

While complement is considered a first line of defense against invading pathogens, including viral infections, blocking C3 activation can significantly attenuate the lung-directed proinflammatory sequelae of infections [116]. Both the genetic absence of C3 and the blockade of downstream complement effectors, such as C5a/C5aR1, have shown therapeutic promise by containing the detrimental proinflammatory consequences of viral spread mainly via inhibition of monocyte/neutrophil activation and immune cell infiltration into the lungs [117]. Furthermore, studies of previous coronaviruses have shown that blocking C3 activation significantly attenuates the lung-directed proinflammatory sequelae of infections [116,118]. A recent study also revealed that coronaviruses’ proteins (SARS-CoV, MERS-CoV, and SARS-CoV-2) bind to a key protein of the lectin pathway (MASP-2/Mannan-binding lectin serine protease 2), leading to complement-mediated inflammatory lung injury. Taken together, several clinical and laboratory data suggest that the complement activation and inhibition need to be further investigated in patients with severe COVID-19 infections [119]. Inflammatory states, such as diabetes and obesity, also activate complement and thus, may exacerbate complement-mediated injury [120]. Patients with severe lung injury may be the most likely to have a genetic predisposition, as well as benefit from complement inhibition.

Given the promising preclinical data and the severity of COVID-19 infections, eculizumab is currently being studied in patients with severe COVID-19 infections (ClinicalTrials.gov Identifier: NCT04288713). In contrast to frequent intravenous infusions every two weeks that are required for eculizumab treatment, a single intravenous dose of ravulizumab should be sufficient in patients with COVID-19. Since C3 and the lectin pathway have been implicated in the pathophysiology of coronaviruses infections, inhibitors of proximal complement pathways, under clinical development for complement-mediated TMAs, could also be efficacious in COVID-19 [119]. Interestingly, the C3 inhibitor AMY-101 has already shown efficacy in COVID-19 [121]. Additional clinical data are needed to provide novel insights in these patients [122].

In summary, a plethora of evidence suggests that complement inhibition could be a promising new approach to contain systemic, complement-mediated inflammatory reactions like CAR-T cell neurotoxicity.

## 6. Conclusions and Future Perspectives

In conclusion, novel cellular therapies have introduced a new era of endothelial injury syndromes. CRS and neurotoxicity present with a different phenotype but share many similarities with the endothelial injury syndromes post HCT (TA-TMA, GVHD or SOS/VOD). The interplay between complement, endothelial dysfunction, hypercoagulability, and inflammation emerges as a common denominator in these syndromes.

Similarly, recent lines of evidence suggest that endothelial dysfunction, hypercoagulability, and inflammation are also key players in the pathophysiology of CAR-T cell toxicity. Since complement inhibition has shown safety and efficacy in patients with endothelial dysfunction syndromes (such as TA-TMA), as well as in patients with excessive inflammation (such as severe COVID-19), it can by hypothesized that complement inhibitors will show efficacy in these new patterns of toxicity.

As the indications for CAR-T cells and the patient populations expand, there in an unmet clinical need of better understanding of the pathophysiology of toxicity following CAR-T cells. Further insights into their pathophysiology will facilitate novel therapeutic options.

## Figures and Tables

**Figure 1 ijms-21-03886-f001:**
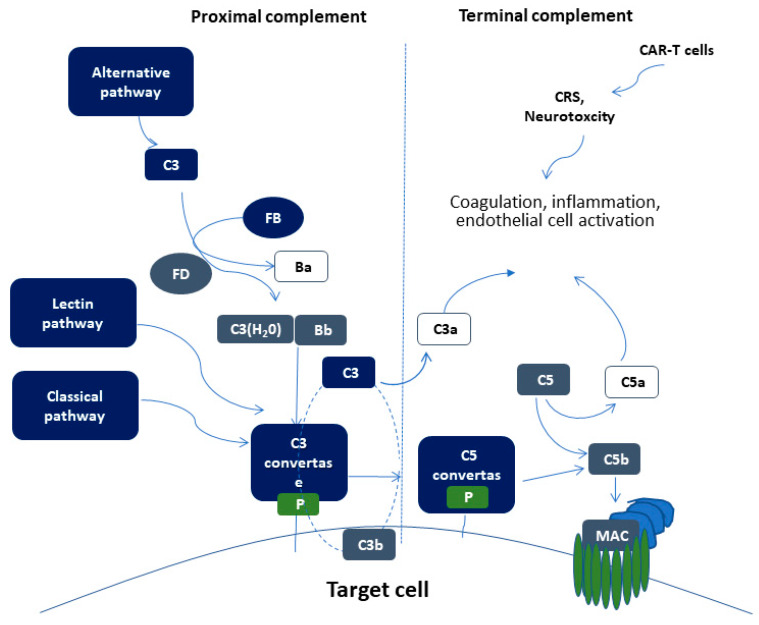
Schematic representation of complement activation. Proximal complement activation initiated by any of the three pathways (classical, alternative, or lectin pathway) leads to C3 activation and C3 convertase formation on C3-opsonized surfaces. C3 activation through the alternative pathway of complement amplifies this response (APC amplification loop, shown in dotted lines), culminating in pronounced C3 fragment deposition on target cells. In the presence of increased surface density of deposited C3b, the terminal (lytic) pathway is triggered, leading to membrane attack complex (MAC) formation on the surface of target cells. C3a and C5a mediate complement interactions with inflammation, coagulation, and endothelial cell activation. These alterations are also triggered by CAR (chimeric antigen receptor)-T cell toxicity syndromes, including CRS (cytokine release syndrome) and neurotoxicity.

**Table 1 ijms-21-03886-t001:** Clinical manifestations of CAR-T cells toxicity.

Toxicity	Manifestations
Cytokine release syndrome (CRS)	fever, hypotension, hypoxia, manifestations from multiple systems: arrhythmia, cardiomyopathy, prolonged QTc, heart block, renal failure, pleural effusions, transaminitis, and coagulopathy
Neurotoxicity	delirium, encephalopathy, somnolence, obtundation, cognitive disturbance, dysphasia, tremor, ataxia, myoclonus, focal motor and sensory defect, seizures, cerebral edema
Cardiotoxicity	sinus tachycardia, hypotension, decreased left ventricular ejection fraction, arrhythmias, QT prolongation, increased troponin
Respiratory toxicity	hypoxia, dyspnea, increased respiratory rate, respiratory failure, pleural effusions, capillary leak syndrome
Hepatic and gastrointestinal toxicity	increased liver transaminases or alkaline phosphatase or direct bilirubin, nausea, vomiting, diarrhea
Hematologic toxicity	anemia, thrombocytopenia, neutropenia, B-cell aplasia, hypogammaglobulinemia, prolongation of partial thromboplastin time (PTT) or prothrombin time (PT), decreased fibrinogen, disseminated intravascular coagulation (DIC), hemophagocytic lymphohistiocytosis
Renal toxicity	renal insufficiency, hyponatremia, hypokalemia, hypophosphatemia, tumor lysis syndrome

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
