# Peer review of "A New Era in Endothelial Injury Syndromes: Toxicity of CAR-T Cells and the Role of Immunity"

_ijms, 2020, doi:10.3390/ijms21113886_

Round 1

Reviewer 1 Report

Gavriilaki et al. expertly review the pathophysiology and therapeutic approaches to complement-associated diseases, subsequently argue that CAR-T-associated toxicities ICANS and CRS both have features of complement-associated disease.

Everything the authors say about complement is competent.

The attempt to tie complement and CARs together lacks supportive evidence and is thus not convincing. Certainly a REVIEW requires that several other groups have worked on a subject, that the review author possesses authoritative knowledge of the subject and can thus assemble the whole picture where the intellectually more limited earlier authors had only presented small pieces of the mosaic with lots of space between them. It is this reviewer’s impression that with respect to CARs and complement this groundwork was not laid and thus the time to review the topic has not come.

What I would encourage the authors, therefore, to do is largely purge the manuscript of CARs and CAR toxicity, to instead dwell on complement and its role in health and disease, possibly develop the topic in slightly more depth, also mechanistically: What does activation of this archaic serine protease cascade do to the body, why is it sometimes good and what happens when it goes overboard, what are the signs, the symptoms, what laboratory tests should be ordered, what complications expected, etc. Complement is, after all, the most ancient piece of our immune system.

In a paragraph “why should you care about complement” it is then certainly reasonable to mention that there are complement-associated illnesses and to list a few and discuss their features. It is fair for the authors to state that they have reason to believe CAR-toxicity should be on the list and why. The list of complement-associated illnesses should be expanded. Many of these have exemplary features that could be explored. Neisseria meningitides sepsis immediately comes to mind and the mitigated cases in patients with hereditary defects in the terminal cascade. I furthermore believe that severe COVID also has a number of features which might be reminiscent of complement activation. If the authors can subscribe to this position and explore it as they have explored the connection of CAR-toxicity and complement (largely theoretical or based on very preliminary evidence), their review might gain additional timeliness.

The authors are exceptionally self-referential. I count at least 18 references to their own work.

Reviewer 2 Report

The review by Gavriilaki et al is describing endothelial injury syndromes in connection to CAR T cells. The topic is contemporary and of great interest to the growing field of CAR therapeutics which includes handling of the adverse reactions. The review is well written and easy to follow. I have minor comments only to improve of this review:

  • Page 2, row 54: Since the authors mention that tisagenlecleucel and axicabtagen ciloleucel are developed by Novartis and Gilead, respectively, it can be good also to mention the same relationship of liso-cel and Bristol-Myers Squibb? Further, state the full INN name prior to the abbreviation liso-cel.
  • Page 3, row 85: The sentence “ These life-threatening syndromes…etc” can be misinterpreted as 93% of all CAR T cell patients have life-threatening adverse events. Please, consider rephrasing.
  • The review very well describes the molecular events upon complement activation and the following cell toxicity, and associates coagulation/complement disorders to cytokine release syndrome. However, the review could benefit from demonstrating the molecular links of cytokines mediating complement activation and hypercoagulation instead of not only claiming their association.

Round 2

Reviewer 1 Report

All comments were ignored, paper is essentially unchanged. Editor to decide.

Author Response

We thank the Editor for providing us the opportunity to improve our manuscript.
1. We have added details on manufacturing and administration of CAR-T cells.
2. We agree with the Editor that providing frequencies would be useful. Unfortunately, these are only clear for CRS and neurotoxicity, still with high range in the three approved products. Regarding the other toxicity, each symptom (such as cough, nausea) has been reported with different frequency (very common, common, or uncommon) in clinical trials. Therefore, it is not possible to provide a frequency for such groups of symptoms, classified per system. We have added relevant data in our manuscript.
3. This line has been deleted since it is explained further in another section.
4. We have incorporated CAR-T cell toxicity in our figure. Since there are no clear data to connect complement and this toxicity, we aimed to describe similarities in the clinical and pathophysiological presentation of such syndromes.